# Placenta-Derived Mesenchymal Stem Cells Restore the Ovary Function in an Ovariectomized Rat Model via an Antioxidant Effect

**DOI:** 10.3390/antiox9070591

**Published:** 2020-07-06

**Authors:** Jin Seok, Hyeri Park, Jong Ho Choi, Ja-Yun Lim, Kyung Gon Kim, Gi Jin Kim

**Affiliations:** 1Department of Biomedical Science, CHA University, Seongnam 13488, Korea; jjin8977@gmail.com (J.S.); hyeyeyeri@gmail.com (H.P.); 2Department of Oral Pathology, Gangneung-Wonju National University, Gangneung 26403, Korea; jhchoi@gwnu.ac.kr; 3Department of Integrated Biomedical and Life Sciences, College of Health Science, Korea University, Seoul 02841, Korea; jayun78@korea.ac.kr; 4Department of Convergence Medicine, School of Medicine, University of Ulsan and Asan Medical Center, Seoul 05505, Korea; kimkyunggon@gmail.com

**Keywords:** premature ovarian failure, ovariectomized rat model, stem cell therapy, placenta-derived mesenchymal stem cells, folliculogenesis, reactive oxidative stress, antioxidants

## Abstract

Oxidative stress is one of the major etiologies of ovarian dysfunction, including premature ovarian failure (POF). Previous reports have demonstrated the therapeutic effects of human placenta-derived mesenchymal stem cells (PD-MSCs) in an ovariectomized rat model (OVX). However, their therapeutic mechanism in oxidative stress has not been reported. Therefore, we investigated to profile the exosome of serum and demonstrate the therapeutic effect of PD-MSCs transplantation for the ovary function. We established an OVX model by ovariectomy and PD-MSCs transplantation was conducted by intravenous injection. Additionally, various factors in the exosome were profiled by LC-MS analysis. As a result, the transplanted PD-MSCs were engrafted into the ovary and the existence of antioxidant factors in the exosome. A decreased expression of oxidative stress markers and increased expression of antioxidant markers were shown in the transplantation (Tx) in comparison to the non-transplantation group (NTx) (* *p* < 0.05). The apoptosis factors were decreased, and ovary function was improved in Tx in comparison to NTx (* *p* < 0.05). These results suggest that transplanted PD-MSCs restore the ovarian function in an OVX model via upregulated antioxidant factors. These findings offer new insights for further understanding of stem cell therapy for reproductive systems.

## 1. Introduction

The ovary is a specialized organ that ensures reproductive success during a definite life stage as the main regulator of female fertility. However, the ovaries are more severely affected by age compared to other organs. Ovarian aging results in ovarian failure and menopause and leads to infertility through the reduction in important sex hormones, follicle loss, and oocyte quality decline [1]. In recent studies, ovarian dysfunction has been shown to increase due to genetic factors (e.g., genomic DNA alteration, mitochondrial DNA mutations, and decreased telomerase activity), as well as various environmental factors (e.g., oxidative stress, advanced glycation, and products) [2]. These processes of ovarian dysfunction accelerate with aging and increase as women age. In particular, POF is the loss of function of the ovaries in women under the age of 40. It is characterized by a decrease in the function of the ovary, which is similar to the menopausal period, but it is different from a general menopausal period. It is defined as a low E2 concentration and high FSH hormone, and it is characterized by various degenerative diseases, including heart disease [3,4,5].

Recently, hormone replacement therapy (HRT) and cryopreservation have been applied to patients as an alternative to overcome ovarian dysfunction. HRT is widely employed as a hormone control method, but it does involve a risk of illness, such as heart disease, stroke, breast cancer, and colorectal cancer, in patients. However, it has recently been reported that for women under the age of 60, the various risks can be reduced when it is used at an appropriate time. However, hormone therapy has the disadvantage of being ineffective when HRT is stopped or not used continuously [6,7]. Ovarian tissue cryopreservation is a new therapy employed to treat women with POF, but it has a lower survival rate of the thawed ovary, including difficulties with conceiving naturally [8]. Therefore, it is important to develop safe new treatments that can recover ovary dysfunction.

PD-MSCs are reported to have multiple abilities, including strong self-renewal, multipotent differentiation, and immunomodulatory properties [9,10]. In addition, PD-MSCs secrete more various cytokines, such as growth factors, compared to other stem cells (e.g., bone marrow-derived MSCs (BM-MSCs) and adipose-derived MSCs (AD-MSCs)). Lee et al. and their colleagues demonstrated that PD-MSCs secrete various cytokines, including G-CSF, RANTES, and IL-6/-8/-10 [11]. Interestingly, several studies have shown that these cytokines also affect the mitochondrial function in cells. Savitree et al. suggested that granulocyte-colony stimulating factors (G-CSF), which are mitochondria-dependent caspase 3 inhibitors, directly protect cardiac mitochondria from oxidative stress [12]. The potential of PD-MSCs has been highlighted in various degenerative diseases due to their therapeutic effects as a new stem cell therapy [13,14,15,16,17,18]. In previous reports, we reported that a 3D spheroid from human PD-MSCs stimulates folliculogenesis, as well as engraftment into the ovary tissues, in an ovarian failure rat model with ovariectomy (OVX) [19].

Exosomes are small extracellular vesicles with a diameter size range of −40 to 160 nm that are secreted from most human cell types and contain a specific composition of protein, lipid, RNA, and DNA [20]. Isolated exosomes, known as markers (such as the tetraspanins family (CD63, CD9, CD81, and CD82) and heat shock proteins (HSPs; HSP60, HSP70, and HSP90) have been characterized [21]. It is proved that exosomes can have various effects in vivo depending on the microenvironment and location of production. Especially, exosomes play a role in cell to cell communication and have been in the spotlight for their clinical applications for various diseases via changed microenvironments [22]. Several researchers have demonstrated that exosomes changed the microenvironment and indicated biomarkers in ovarian cancer, as well as other severe diseases [23,24]. It is reported that secreted exosome by oxidative stress prevents cell death for defense mechanisms [25]. Hence, many studies have been reported on mechanisms using exosomes from stem cells in diseases. However, the study is still insufficient to determine whether the stem cells affect the microenvironment, including exosomes.

Reactive oxygen species (ROS), such as molecules of superoxide, hydroxyl radicals, hydrogen peroxide, and hypohalous acids, are generated by H_2_O_2_ and the NADPH oxidase system during cellular metabolism and removed by antioxidants such as superoxide dismutase, glutathione peroxidase (GPX), and catalase (CAT) [26,27]. Moreover, it is known that oxidative stress is increased in damaged cells/tissues and induces damage. At this time, the mechanism for deciphering the reactive intermediates is activated to recover and regenerate the damaged cells and maintain homeostasis of the body [28,29]. A previous study demonstrated that ROS activate p53 and upregulate the expression level of the pro-apoptotic proteins of BAX and PUMA, resulting in cell apoptosis [30]. In the ovary, ROS are generated during follicular rupture, which is an inflammation-like reaction, and oocyte ovulation [31]. However, excessively accumulated ROS trigger infertility by a loss of oocyte maturation and granulosa cell luteinization [32,33]. Several recent studies have reported that increases in the ROS level reduce the oocyte quality, reproductive outcome, and granulosa cell apoptosis [34,35]. However, because there is still a lack of research, understanding and studying the correlation between oxidative stress and ovarian dysfunctional disorder is necessary. Therefore, in this study, we aimed to investigate the therapeutic effect of PD-MSCs via antioxidant signaling in an OVX rat model.

## 2. Materials and Methods

### 2.1. Animals

All experiments involving animals were performed in accordance with the animal care guidelines issued by the National Institutes of Health and were approved by the Institute Animal Care and Committee of the CHA Laboratory Animal Research Center at Sampyeng-dong in Gyeonggi, Korea (IACUC 190048). All female Sprague-Dawley rats were purchased from Orient Bio Inc. (Orient Bio Inc., Seongnam, Gyeonggi, Korea) and were 8 weeks old in this study. The rats were housed in groups of two rats per plastic cage with corn-cob bedding and were provided with ad libitum access to standard commercial food and tap water. The temperature was 21 °C, and a 12 h/12 h light-dark cycle was employed.

### 2.2. Ovariectomized Rat Model Establishment

All animal experiments were approved by the Institutional Animal Care and Use Committee (IACUC 190048) of the CHA Laboratory Animal Research Center (Gyeonggi-do, Korea). The following week, 40 animals were randomly allocated into two groups. The NTx group included ovariectomized (OVX) rats (*n* = 20), and the Tx group included indirectly stem cell transplanted OVX rats obtained through a tail vein injection (*n* = 20). Ovariectomy was performed in female rats of all groups to remove one of the ovaries. All rats were anesthetized via intraperitoneal injection with 250 mg/kg avertin (Sigma-Aldrich, St. Louis, MO, USA). After all rats had been sterilized using 70% ethanol with distilled water, the skin and muscles in the pelvic area of the back were incised and the tissue of one ovary was tied off with a sterile suture and removed. After removal of the ovary, the surgical site was disinfected with povidone-iodine (Sigma-Aldrich, St. Louis, MO, USA) and all OVX rats were maintained in their housing cages for one week.

### 2.3. Cell Culture of PD-MSCs and Transplantation into an Ovariectomized Rat Model

Placentas were collected from women who were free of any medical, obstetrical, or surgical complications and who delivered at term (38 ± 2 gestational weeks). PD-MSCs were isolated from human placental chorionic plates and approved by the Institutional Review Board of CHA General Hospital, Seoul, Korea (IRB 07-18). PD-MSCs were isolated from chorionic plates of normal-term placentas, as previously described by Lee et al. [13]. Briefly, PD-MSCs were cultured in alpha-minimum essential medium (α-MEM; Hyclone, GE healthcare life sciences, Seoul, Korea) supplemented with 10% fetal bovine serum (FBS; Gibco-BRL, Rockville, MD, USA), 1% penicillin/streptomycin (Pen-Strep; Gibco-BRL), 25 μg/mL human fibroblast growth factor 4 (hFGF-4; Peprotech Inc., Rocky Hill, NJ, USA), and 1 μg/mL heparin (Sigma-Aldrich) at 37 °C in an incubator with a humidified atmosphere of 5% CO_2_. One week after the ovariectomy, PD-MSCs (5 × 10^5^) were labeled using a PKH67 Fluorescent Cell linker kit (Sigma-Aldrich) and injected through the tail vein. After blood samples had been collected for hormone level analysis, the rats of all groups were sacrificed, and ovary tissues were harvested at 1, 2, 3, and 5 weeks using liquid nitrogen. All ovary tissues and blood samples of each group (NTx and Tx; 1, 2, 3, and 5 weeks; *n* = 5) were pooled to ensure there was no variation between the groups.

### 2.4. Exosome Sample Preparation for Proteome Analysis

To analyze the exosome of the serum in the OVX rat model, we isolated the exosome using a precipitation kit (System Biosciences, Palo Alto, CA, USA), following the manufacturer’s instructions. Protein amounts of the isolated exosome samples were measured using a bicinchoninic acid (BCA) assay and 100 μg of each protein was taken and dried. Each sample was lysed in 300 μL of lysis buffer consisting of 5% sodium dodecyl sulfate and 50 mM triethylammonium bicarbonate (pH 7.55, Thermo Fisher Scientific, Waltham, MA, USA) by sonication on ice. The lysates were cleared by centrifugation at 15,000 rpm for 15 min at 4 °C. Each sample underwent STrap-based tryptic digestion employing previously known methods [36] using a trypsin/LysC mixture (Promega, Madison, WV, USA).

### 2.5. Nano-LC-ESI-MS/MS Analysis

Samples were analyzed on a Dionex UltiMate 3000 RSLC nano LC system (Thermo Scientific, Waltham, MA, USA) coupled to a Q Exactive plus mass spectrometer (Thermo Scientific) with a nano-ESI source. Tryptic peptides from a bead column were reconstituted using 0.1% formic acid and were loaded via an Acclaim PepMap 100 trap column (100 μm × 2 cm, nanoViper, C18, 5 μm, 100 Å, Thermo Scientific). Subsequent peptide separation was performed on an Acclaim PepMap rapid separation LC (RSLC) analytical column (75 μm × 50 cm, nanoViper, C18, 2 μm, 100 Å, Thermo Scientific) for over 200 min (250 nL/min) using a 0% to 40% acetonitrile gradient in 0.1% formic acid at 50 °C. Mass spectra were acquired in a data-dependent mode with automatic switching between a full scan (*m*/*z* 350–1800) and 20 data-dependent MS/MS scans. The target value for the full-scan MS spectra was 3,000,000, with a maximum injection time of 100 ms and a resolution of 70,000 at *m*/*z* 400. The ion target value for MS/MS was set to 100,000 with a maximum injection time of 50 ms and a resolution of 17,500 at *m*/*z* 400 with normalized collision energy (27%). The dynamic exclusion of repeated peptides was applied for 20 s. For each biological sample, three technical replicates were performed.

### 2.6. Database Search and Label-Free Quantitation and staTistical Analysis

The acquired MS/MS spectra were searched using the SequestHT on Proteome discoverer (version 2.2, Thermo Fisher Scientific) against the SwissProt database (June 2019). Briefly, the precursor mass tolerance was set to ±10 ppm and the MS/MS tolerance was set to 0.02 Da. The search parameters were set as default, including cysteine carbamidomethylation (CAM) as a fixed modification, and N-terminal/lysine acetylation, N-terminal methionine loss, and acetylation, methionine oxidation, phosphoserine, phosphor-threonine, and phosphor-tyrosine as variable modifications with two miscleavages. False discovery rates (FDRs) were set to 1% for each analysis using “Percolator”. From the SequestHT search output, the peptide filters for peptide confidence, peptide rank, score versus charge state, and search engine rank were applied at the default values of the Proteome discoverer. Label-free quantitation was performed using the peak intensity for unique and razor peptides of each protein. Normalization was conducted using the total peptide amount. For differential analysis of the relative abundance of proteins between samples, the free software Perseus (version 1.6.10.43; Maxquant, Germany) was used. The values of normalized protein abundances were transformed into log2 scale values. Technical replicates of each sample were grouped and a minimum of three valid values was required in at least one group. To find statistically significant differences between samples, a t-test was performed using permutation-based FDR (0.05 cut-off). The hierarchical clustering of enrichment z-score values was performed using InstantClue software (http://www.instantclue.uni-koeln.de) [37]. Hierarchical clustering was performed using the Euclidean distance as a metric and the average linkage. The quantitation data were uploaded into the Ingenuity Pathways Analysis (IPA; Ingenuity Systems, Redwood City, CA, USA) program. Data were analyzed through the Ingenuity Pathway Analysis platform (QIAGEN Inc., Valencia, CA, USA). Proteins in the proteomic data were mapped to corresponding gene objects in the Ingenuity Pathways knowledge base. Then, biological networks were generated using the knowledge base for interactions between the uploaded gene list and all other gene objects stored in the knowledge base. A functional analysis of the networks was conducted to identify the biological functions and/or diseases that were most significant to the genes in the network. The score was derived from the p-value of the test and indicated the likelihood of the mapped genes in a network being found together due to random chance (score = −log_10_p).

### 2.7. Hormone ELISA Assay

All blood samples were collected from the aorta in rats of NTx and Tx groups via the recto-orbital technique. The serum samples were separated from whole blood by using a blood collection tube (vacutainer; BD Biosciences, San Jose, CA, USA) at 1300 RCF for 15 min. All blood serum was stored at −80 °C and the estrogen (Bio vision, Milpitas, CA, USA), anti-Mullerian hormone (AMH; Elabscience Biotechnology, MA, USA), follicle-stimulating hormone (FSH; Abnova, Taipei, Taiwan), and active caspase-3 (Mybiosource, San Diego, CA, USA) activity in serum were analyzed by ELISA kits, following the manufacturer’s instructions. In brief, an equal volume of sample was added into the specific antibody-coated plates. Next, the specific horseradish peroxidase (HRP)-conjugates were added to each well and incubated at 37 °C. After the substrates had been added and incubated in the dark for substrate development, the antibody activity was analyzed by using a microplate reader (BioTek, Winooski, VT, USA). All samples were tested three times and the results are presented as the relative value.

### 2.8. Genomic DNA Isolation

To conduct genomic DNA (gDNA) isolation, the ovary tissues of rats were homogenized in LN2 and isolated by proteinase K (Qiagen) and lysis buffer, including radioimmunoprecipitation assay buffer (RIPA) buffer (Sigma-Aldrich), phosphatase inhibitor (Roche Diagnostics, Manheim, Germany), and protease inhibitor (A.G. Scientific, San Diego, CA, USA), for 4 h at 55 °C. Next, phenol:chloroform:isoamyl alcohol (25:24:1; Sigma-Aldrich) was added to ligated samples and collected by centrifugation at 13,000 rpm for 30 min. After this processing had been repeated twice, the digestion buffer containing iso-amylalchol (Sigma-Aldrich) and 0.3 M sodium acetate were added into the samples and incubated at −20 °C overnight. After this, the pellet of gDNA was washed with 70% cold ethanol and eluted using Tris-EDTA buffer. The gDNA concentration was measured by nanodrop analysis. The gDNA was analyzed by human-specific primers. Primer sequences for human-specific *Alu* sequences (Accession number U14573) were as follows: (Forward) 5′-GGA GGC TGA GGC AGG AGA A-3′ and (Reverse) 5′-ATC TCG GCT CAC TGC AAC CT-3′. The relative mRNA expression of the human Alu sequence gene was normalized by the cycle threshold values of rat *GAPDH*.

### 2.9. RNA Isolation and Quantitative Real-Time Polymerase Chain Reaction Analysis

Total RNA was isolated from snap-frozen ovaries using TRIzol reagent (Ambion, Boston, MA, USA), according to the manufacturer’s protocols. The isolated RNA concentration was quantified by using a nanodrop spectrophotometer (Thermo Scientific). A total of 1 μg of RNA was reverse transcribed into cDNA using the Superscript kit (Invitrogen, Carlsbad, CA, USA) containing 20 pmol of oligodT, 10 mM of DNTP mix, RNase out, and Superscript III reverse transcriptase. The cDNA synthesis conditions were 65 °C for 5 min, 50 °C for 1 h, and 72 °C for 15 min. Amplification of the cDNA was accomplished in triplicate using a real-time polymerase chain reaction (PCR; Bio-Rad Laboratories Inc., Berkeley, CA, USA) instrument in the presence of commercially available SYBR Green PRC Master mix (Roche). The mRNA amplification conditions included incubation at 95 °C for 5 min, followed by 40 cycles at 95 °C for 5 sec and 60 °C for 30 sec. All reactions were performed in triplicate. Alu sequences (Accession number U14573) were as follows: (Forward) 5′-GGA GGC TGA GGC AGG AGA A-3′ and (Reverse) 5′-ATC TCG GCT CAC TGC AAC CT-3′. The relative mRNA expression of the human Alu sequence gene was normalized by the cycle threshold values of rat GAPDH.

### 2.10. Protein Isolation and Western Blot Analysis

All tissues were pooled to ensure no variation between groups. The ovary tissues were lysed using a lysis buffer containing protease inhibitor (Roche) and phosphatase inhibitor (A.G. Scientific., Sandiego, CA, USA) with RIPA buffer (Sigma-Aldrich) and then centrifuged at 13,000 g for 15 min at 4 °C. The supernatants were collected for the next western blot. The concentration of proteins was determined with BSA kits (Thermo Fisher). Next, the total amount of proteins was separated by 6-15% sodium dodecyl sulfate-polyacrylamide gel electrophoresis (SDS-PAGE) and transferred onto polyvinylidene difluoride membranes (PVDF; Bio-Rad). After being blocked with 4% (*w*/*v*) bovine serum albumin (BSA; Amresco, Solon, OH, USA) in Tris-buffered saline (Elbio, Deajun, Korea) containing 0.1% Tween-20 (Elbio) for 1 h at room temperature, the PVDF membranes were incubated with the primary antibody overnight at 4 °C. Primary antibodies included the following: *P4HB* (1:1000; Abcam, UK Cambridge), *HO-1* (1:500; Novus Biologicals, CO, USA), *HO-2* (1:500; Novus), *Prdx1* (1:500; Abcam), *Prdx2* (1:1000; Abcam), *catalase* (1:1000; Abcam), *SOD1* (1:500; Cell Signaling Technology, MA, USA), *Lhx8* (1:500; Santacruz, Texas, USA), *Nobox* (1:1000, Santacruz), Nanos3 (1:500; Abcam), and *GAPDH* (1:5000; Abfrontier, Seoul, Korea). After incubation in the primary antibodies, membranes were washed with tris buffered saline-tween 20 (TBS-T) for 5 × 3 min and incubated with the secondary antibody (Rabbit: 1:20,000; Mouse: 1:10,000; Goat: 1:5000) for 1 h at room temperature. After washing the samples with TBS-T buffer for 5 × 3 min at room temperature, the immunoreactivity was visualized by enhanced chemiluminescence using enhanced chemiluminescence (ECL) kits (Bio-Rad) and Chemi-doc (Bio-Rad).

### 2.11. Ovarian Follicle Counting Using Hematoxylin and Eosin (H&E) Staining

All ovary tissues were fixed with 10% neutral buffered formalin solution containing 1× phosphate-buffered saline (PBS), 37% formaldehyde (Merck, Darmstadt, Germany), and distilled water overnight. The ovary tissues were dehydrated, embedded in paraffin, and cut into sections (4–5 μm). To analyze the formation of ovarian follicles in the ovaries, the tissue section slide was stained with hematoxylin (DAKO, Carpinteria, CA, USA) and eosin. Images were captured by using a Nikon microscope at ×1 and ×40 magnifications. The follicles were counted in two sections, which were the total follicles, including primordial, primary, secondary, and preovulatory follicles, and the antral follicles.

### 2.12. PKH67-Labeled PD-MSCs Tracking

To analyze the location of transplanted PD-MSCs in the ovary, PD-MSCs with labeled PKH67 were transplanted into the OVX rat model. After this, the ovarian frozen block was cut into sections (6–7 μm). After ovary tissues had been washed with 1× phosphate-buffered saline (PBS) buffer, sectioned ovary tissues were incubated with propidium iodide (PI; SIGMA) solution for 30 min at 37 °C. Next, the sectioned ovary tissues were washed with 1× PBS and mounted using a mounting solution (DAKO). The fluorescence signal was observed by confocal instruments. The images were obtained from three independent trials.

### 2.13. Immunofluorescence Staining

To analyze the superoxidase in the mitochondria of ovarian follicles, the ovarian frozen block was cut into sections (6–7 μm). After the ovary tissues had been washed with hanks’ balanced salt solution (HBSS; GIBCO-BRL) buffer, they were incubated with 3 μM Mito SOX (Invitrogen, Carlsbad, CA, USA) and 100 nM Mito Tracker (Invitrogen) for 40 min at 37 °C. Next, the ovary tissues were washed with HBSS and incubated with 0.5 μg 4′,6-diamidino-2-phenylindole (DAPI; SIGAM-Aldrich) for 1 min at room temperature. After the reaction, the ovary tissues were washed and mounted by using a mounting solution (DAKO). The detection of superoxide (red fluorescence) and mitochondria (green fluorescence) was conducted using confocal instruments. All data and images were obtained from three independent trials.

### 2.14. Immunohistochemistry

Ovary tissues were fixed by 10% neutral buffered formalin (BBC Chemical, Mount Vernon, WA, USA), embedded in paraffin, and sectioned with 6 μm of the ovaries. Sectioned ovary tissues were deparaffinized in a 60 °C dry oven by xylene and ethanol. Deparaffinized tissues underwent antigen retrieval by an EDTA (eLbio, Seongnam-Si, Korea) reaction. After removing the unbound primary antibody, the tissues in slides were incubated with DAKO Real EnVision HRP Rabbit/Mouse secondary antibody (Dako) at room temperature for 1 h. The slides were incubated with DAB and counterstained with hematoxylin (Dako). After the reaction, slides were rinsed with tap water. Slides underwent dehydration by ethanol and xylene. Tissues were analyzed by the 3D HISTECH program (The Digital Pathology Company, Budapest, Hungary).

### 2.15. Statistical Analysis

For each set of data, independent experiments were repeated at least three times, with data representing the mean ± standard deviation of at least three experiments. The statistical differences between experimental groups were determined by a t-test. The results were considered significant at *p* < 0.05.

## 3. Results

### 3.1. Transplanted PD-MSCs Engraft into the Damaged Ovary of an Ovariectomized Rat Model

As Figure 1A shows, the PKH67-labeled PD-MSCs were located near the follicles of Tx groups at 1 week (Figure 1A). The human *Alu*-sequence gDNA expression was analyzed in the OVX group. The gDNA expression of the *hAlu*-sequence gene indicates the engraftment of human PD-MSCs. Interestingly, the gDNA expression of the *hAlu*-sequence was remarkably increased at 1 week after PD-MSCs in the Tx group compared to the NTx group. However, human Alu-sequence expression was not detected in the NTx group (*p* < 0.05; Figure 1B). The mRNA expression of the human Alu-sequence was significantly increased at 1 and 3 weeks after PD-MSCs in the Tx group compared to the NTx group. The mRNA expression of the human Alu-sequence demonstrates the metabolic activity of the engrafted human PD-MSCs (*p* < 0.05; Figure 1C). These results indicated the homing activity of PD-MSCs to the ovary in the OVX model after PD-MSCs in the Tx group.

### 3.2. The Analysis of Exosomes from Serum of the OVX Rat Model and the Expression of Antioxidant Factors

To confirm the alteration of exosomes, we conducted an exosome analysis of the blood of the OVX rat model according to PD-MSCs Tx. As shown in Figure 2A,B, we identified the consequences of changing the exosomal components in the blood as stem cells are transplanted into the OVX rat model (Figure 2A,B). Interestingly, the exosome components in the blood exhibited changes in the expression of mitochondrial antioxidant factors, including energy metabolism and hormone and follicle development after PD-MSCs transplantation into the OVX rat model. Therefore, to analyze the antioxidant effect of mitochondria in the OVX rat model according to PD-MSCs Tx, we selected and compared the antioxidant factors among various factors and conducted the analysis at a protein level. Firstly, protein disulfide-isomerase (*P4HB*), which is well-known as a molecular chaperone that generates ROS, was decreased in Tx compared to NTx by label-free quantitation using high-resolution LC-MS and western blot analyses (Figure 2C,D, Appendix A). Secondly, *catalase*, which is an antioxidant enzyme that protects cells from oxidative stress, was significantly increased in Tx compared to NTx (Figure 2E,F, Appendix A). *Peredoxin1* and *Peredoxin2* (*Prdx1* and *Prdx2*), which are antioxidant enzymes that prevent oxidative damage, were remarkably increased in Tx compared to NTx (Figure 2G–J, Appendix A). These results indicated that transplanted PD-MSCs changed the microenvironment of the OVX rat model via the upregulated expression of various antioxidant enzymes.

### 3.3. PD-MSCs Transplantation Reduced Oxidative sTress and Enhanced the Antioxidant Effect in an Ovariectomized Rat Model

As shown in Figure 3, the fluorescence intensities of MitoSOX in the ovary of the NTx group were higher compared to those in the Tx group. Those of the ovary of the Tx group exhibited the lowest accumulation of superoxide in the mitochondria and oocyte (Figure 3A). As a result of measuring the ratio between MitoSOX and MitoTracker, the ratio of MitoSOX/MitoTracker of antral follicles in the NTx group was shown to be remarkably increased compared to in the normal and Tx groups. In addition, the ratio of MitoSOX/ MitoTracker of the Tx group exhibited a lower expression in antral follicles compared to the NTx group (*p* < 0.05, Figure 3B).

Next, we analyzed heme oxygenase-1 and -2 (*HO-1* and *HO-2*) protein expression in the ovary of OVX groups according to PD-MSCs transplantation. The expressions of *HO-1* and *HO-2* genes were induced by oxidative stress. The gene expression of *HO-1* in Tx groups was increased compared to that in NTx groups. The gene expression of *HO-2* was significantly increased at 1 and 2 weeks after PD-MSCs transplantation compared to in the NTx group. However, *HO-2* protein expression was significantly decreased at 3 and 5 weeks after PD-MSCs transplantation compared to that of the NTx group (*p* < 0.05; Figure 3C).

The protein expression of superoxide dismutase (*SOD1*), which is an enzyme that alternately catalyzes the dismutation of superoxide radicals, was analyzed in the OVX model. The gene expression of *SOD1* was significantly increased in 1 week in the Tx group compared to the NTx group (*p* < 0.05, Figure 3D). The gene expression of *catalase*, which is related to protecting the cell from oxidative damage by ROS, was investigated in the OVX model. The gene expression of *catalase* was significantly increased after PD-MSCs in Tx groups compared to that of NTx groups (*p* < 0.05, Figure 3D). These results suggested that PD-MSCs transplantation inhibited ROS production in mitochondria in ovary tissues through enhanced antioxidant factors in the OVX rat model.

Generally, it is important to maintain the balance between proliferation and cell death in organ development, as well as function. Therefore, we analyzed the proliferating cell nuclear antigen (*PCNA*) expression related to proliferation in follicles of the ovary of the rat, regardless of PD-MSCs transplantation. As shown in Figure 4A, *PCNA* positive signals in follicles were remarkably decreased in the NTx group compared to the control. However, *PCNA* positive signals in follicles of the Tx group were increased compared to those of the NTx group (Figure 4A). The results of the ELISA assay demonstrated that the activated *caspase-3* activity was decreased after PD-MSCs in the Tx group compared to NTx groups. In particular, the activated *caspase-3* activity was significantly decreased at 1 and 3 weeks after PD-MSCs in Tx compared to the NTx groups (*p* < 0.05, Figure 4B). The gene expression of *Bcl2* in the Tx group showed the tendency to increase at 1 and 3 weeks compared to that of the NTx group (Figure 4C). The gene expression of *cytochrome C* in the Tx group was significantly decreased at 3 and 5 weeks compared to that of the NTx group (*p* < 0.05, Figure 4D). These results suggest that PD-MSCs induce anti-apoptotic events in ovary tissues of the OVX model.

### 3.4. PD-MSCs Transplantation Enhanced the Ovarian Function through Follicular Development and Hormone Expression in an Ovariectomized Rat Model

The ovarian function can be predicted by follicular development and hormonal changes. Therefore, we analyzed the hormone levels of anti-Mullerian hormone (AMH) and estrogen (E2), which are representative predictive markers for the ovarian function, in the blood of the OVX model, regardless of PD-MSCs transplantation, using ELISA assay kits. In particular, the AMH levels of serum, which have key roles in follicular growth and folliculogenesis, were decreased in the NTx group compared to normal groups after ovariectomy. Interestingly, the AMH levels of serum were significantly increased in the Tx group at 5 weeks after PD-MSCs transplantation compared to those of the NTx group (*p* < 0.05, Figure 5A). Additionally, the estrogen (E2) levels of serum, which included the release of luteinizing hormone, were decreased in the NTx group compared to normal groups after ovariectomy. However, the E2 levels of serum were significantly increased in the Tx group compared to normal groups after PD-MSCs transplantation (*p* < 0.05, Figure 5B).

To analyze the factors related to follicle development in the ovary of the OVX model, we analyzed the expression of the makers related to follicle development after PD-MSCs transplantation. The newborn ovary homeobox (*Nobox*) protein expression, which was expressed in the oocyte, was increased after PD-MSCs transplantation compared to NTx groups. The expression of LIM homeobox protein 8 (*Lhx8*), which is an essential transcription factor of oogenesis processing, was increased in the Tx group compared to the NTx group. The expression of the Nanos homolog 3 (*Nanos3*) protein related to germ cell development was significantly increased at 1, 2, and 3 weeks after PD-MSCs transplantation compared to the NTx group (*p* < 0.05, Figure 5C and Appendix A). These results indicate that PD-MSCs transplantation can improve hormone levels (e.g., AMH and E2) and follicular growth in a rat model with ovariectomy.

### 3.5. PD-MSC Transplantation Improved Follicular Development in an Ovariectomized Rat Model

We confirmed the number of follicles in the OVX model after PD-MSCs transplantation. As Figure 6A shows, the number of follicles in the ovary was increased in all Tx groups (Figure 6A). To quantify the follicles according to each stage of maturity, primordial to antral follicles were measured by quoting Myers et al. [38]. The total follicles in NTx groups were significantly decreased compared to the normal group after ovariectomy. In particular, the number of primordial follicles compared to antral follicles was significantly decreased in the NTx group compared to those of the normal group; otherwise, the total follicles in the Tx group were remarkably increased (*, ** *p* < 0.05, Figure 6B). The number of primordial follicles compared to antral follicles was extremely increased in the Tx group compared to those of the NTx group (*, ** *p* < 0.05, Figure 6B). These results indicate that PD-MSCs transplantation improves the total number of follicles decreased by ovariectomy and maturates the primordial follicles to antral follicles.

## 4. Discussion

POF appears in women aged 40–50 years and is characterized by sex hormone deficiency, a loss of follicular activity, increasing atretic follicles, and infertility [39]. These risk factors are still not sufficiently known, although several studies have reported a correlation between risk factors and ovary dysfunction. Therefore, in this study, we focused on the efficacy of PD-MSCs in follicular development in an OVX rat model through antioxidant effects.

In recent years, the therapeutic effect of stem cells has been reported in degenerative diseases, including ovary dysfunction. Several previous studies have demonstrated that various stem cells improved the ovarian function, including folliculogenesis, sex hormone levels, and the ovarian niche [40,41,42]. BM-MSCs and amniotic epithelial cells (AECs) in an ovary dysfunction model inhibited apoptosis signaling in granulosa cells and restored the ovary function in chemotherapy-induced models [43,44]. Li et al. reported that human chorionic plate-derived MSCs (CP-MSCs) isolated from a normal-term placenta restored the ovarian function in a cyclophosphamide-induced POF mouse model [45]. Liu et al. and their colleagues demonstrated that BM-MSCs have an effect of homing and restored the function in an ovary dysfunctional model [46]. In our previous study, we demonstrated that an engrafted 3D spheroid type of PD-MSCs improved the ovarian function via upregulated gene expression related to folliculogenesis, as well as maturation follicles, in an OVX rat model [19]. These reports suggest that engrafted MSCs enhance the ovarian function through increasing the expressions of target genes related to folliculogenesis, as well as anti-apoptosis and the homing activity of MSCs. In this study, we also confirmed that PKH67-labeled PD-MSCs engrafted into ovary tissues of the OVX rat model after the intravenous transplantation of PD-MSCs, as well as hAlu expression in mRNA of ovary tissues (Figure 1). Therefore, this data demonstrates that transplanted PD-MSCs display homing activity in an ovary injured by ovariectomy through intravenous transplantation. This ability of MSCs is one of their major characteristics and correlates with their therapeutic efficacy in degenerative diseases. We confirmed that transplanted PD-MSCs trigger an upregulated gene expression related to folliculogenesis and hormone levels (e.g., AMH and E2) in the ovary of the OVX rat model (Figure 5 and Figure 6). This data also indicates that the transplanted PD-MSCs improved ovarian function through antioxidants effects and anti-apoptosis.

The exosome can modulate the organ function of dysfunctional disease because it can produce passive movement in body fluids. Previous reports suggest that exosomes regulate the function of the ovary through cell to cell communication [47]. Hence, many researchers have reported that exosomes derived from MSCs have treatment potentials. Han et al. demonstrated that exosomes suppress apoptosis, including *caspase-3* activation, via reduced mitochondrial membrane potential loss [48]. Sun et al. reported that exosomes derived from UC-MSCs suppress stress and apoptosis of granulosa cells by cisplatin [49]. Recently, it has been reported that exosomes secreted from ROS-exposed cells include antioxidant components for the defense of oxidative stress and also trigger tolerance to oxidative stress [25,50,51]. However, the study of microenvironmental change by MSCs is still not understood in ovarian dysfunctional models. Therefore, we confirmed whether the microenvironment in vivo is controlled by PD-MSCs transplantation. Firstly, we performed exosome profiling by LC-MS in the blood of OVX rat models after PD-MSCs transplantation. As results, the various antioxidants factors in exosomes from serum were found in serum after PD-MSCs transplantation (Figure 2). Among the several components in exosomes, selected antioxidant factors (e.g., *catalase*, and *Prdx-1/-2*) were increased by PD-MSCs transplantation. The expression of *catalase* was also increased in blood as well as ovary tissues after PD-MSCs transplantation. The selected factors are involved in the mitochondrial function and protect cell death through decreasing the ROS level in mitochondria of follicles in ovary tissues [52,53].

Oxidative stress-induced ROS result in cellular apoptosis in granulosa cells [54,55]. Several studies have reported a correlation between ROS levels and ovary dysfunction [56]. Shen et al. and their colleague reported that oxidative stress-induced apoptosis is associated with ovarian dysfunction by granulosa cell death. Furthermore, oxidative stress-induced apoptosis causes follicular atresia through Foxo1 overexpression [57]. Lee et al. demonstrated that the gene expression of *HO-1* is increased by depleted glutathione in the ovary and induced oxidative stress by deficient estrogen levels in ovariectomized rat models [58]. In our study, the ROS levels were remarkably decreased in the Tx group compared to those of the NTx group through *HO-1/-2* gene expression. Antioxidant markers (e.g., *SOD1* and *catalase*) were significantly increased in the Tx group compared to the NTx group (Figure 3). Therefore, these previous reports are well-matched with our results and support our data. Finally, PD-MSCs regulate the microenvironment for protect oxidative stress in injured ovary tissues and restore the ovarian function in an OVX rat model through follicular development by antioxidants and an anti-apoptotic effect.

## 5. Conclusions

The transplanted PD-MSCs migrated to the follicular periphery of the damaged ovarian tissue and improved the antioxidant efficacy by altering the *HO-1*/*HO-2* expression in ovarian tissues and by increasing *SOD1* and *catalase* gene expression. These results suggest that PD-MSCs transplanted into ovarian dysfunctional animal models reduce oxidative stress and apoptosis in injured ovarian tissue. Moreover, engrafted PD-MSCs trigger a restored ovary function, including increased gene expression related to follicle development, E2 and FSH hormone levels, and engagement in follicular maturation. These findings offer new insights for further understanding of stem cell therapy for reproductive systems and should provide new avenues to develop more efficient therapies in degenerative medicine.

## Figures and Tables

**Figure 1 antioxidants-09-00591-f001:**
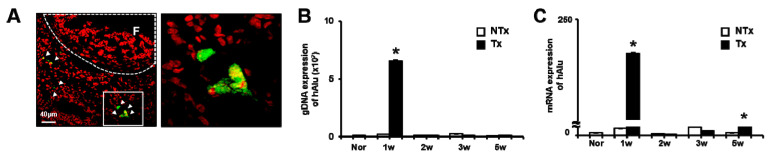
Placenta-derived mesenchymal stem cells (PD-MSCs) engrafted into the ovary of an ovariectomized rat model (OVX) rat model. The expression of PKH67-labeled PD-MSCs was localized in 1 week in the transplantation group (Tx) (**A**). The gDNA and mRNA expression levels of human Alu sequences in ovary tissues of the non-transplantation group (NTx) and Tx were analyzed by qRT-PCR (**B**,**C**). The data are representative of three independent experiments and expressed as means ± SD. * *p* < 0.05: NTx vs. Tx.

**Figure 2 antioxidants-09-00591-f002:**
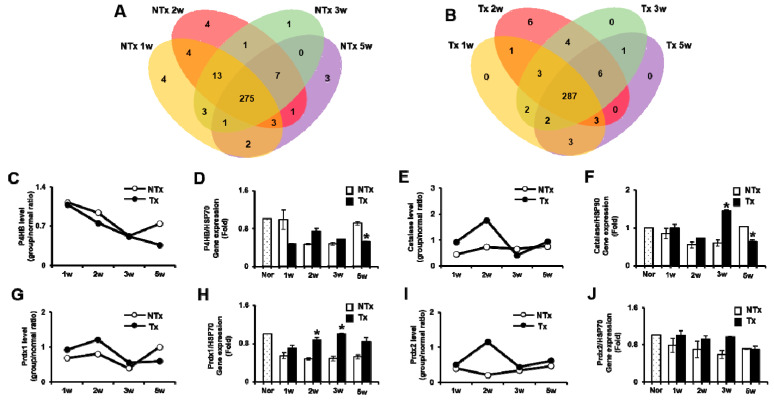
PD-MSCs trigger the upregulated expression of antioxidant enzymes in the exosome of the OVX rat model. The differences of factors in exosomes as a Venn diagram (**A**,**B**). The *P4HB* expression related to the reactive oxygen species (ROS) level was analyzed in the exosome of serum (**C**,**D**). The *catalase* (**E**,**F**), *Prdx1* (**G**,**H**), and *Prdx2* (**I**,**J**) expressions related to antioxidants were analyzed in the exosome of serum. The data are representative of three independent experiments and expressed as means ± SD. * *p* < 0.05: NTx vs. Tx.

**Figure 3 antioxidants-09-00591-f003:**
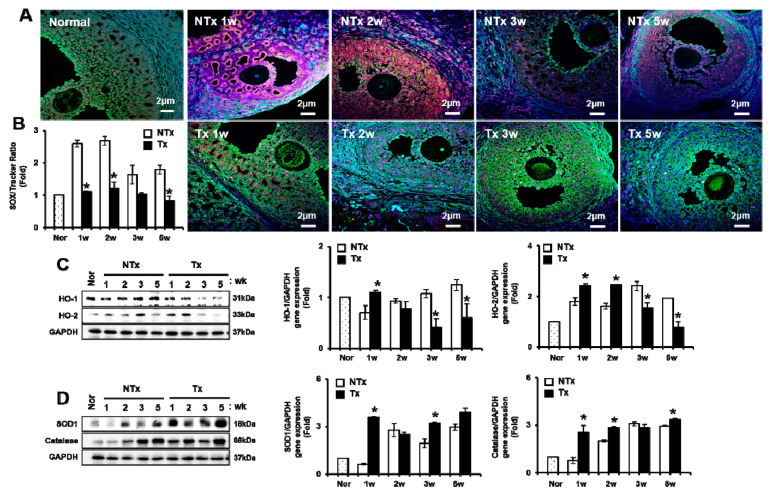
Effect of PD-MSCs transplantation on antioxidants in ovary tissues of the OVX rat model. The superoxide (Mito SOX) and mitochondria (Mito Tracker) in the ovary were stained by immunofluorescence (**A**) and measured by the Image J program (**B**). *HO-1* and *HO-2* gene expression related to oxidative induced factors (**C**) and *SOD1* and *catalase* gene expression related to antioxidant factors (**D**) were analyzed by western blot. The data are representative of three independent experiments and expressed as means ± SD. * *p* < 0.05: NTx vs. Tx3.4. PD-MSCs Transplantation Inhibited Apoptosis in an Ovariectomized Rat Model

**Figure 4 antioxidants-09-00591-f004:**
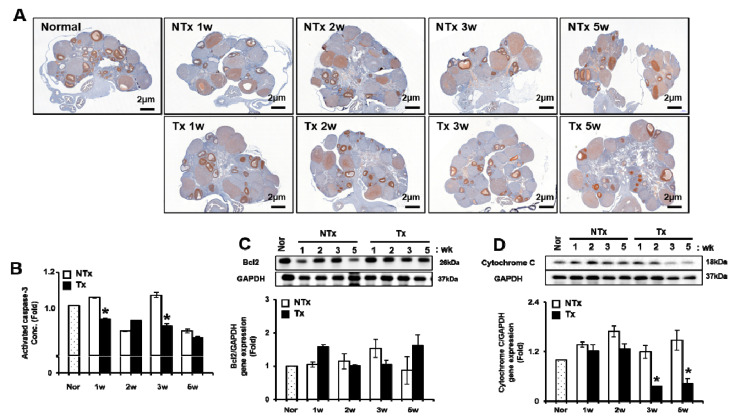
Effect of PD-MSCs transplantation on apoptosis in the ovary tissues of the OVX rat model. The *PCNA* expression in follicles of the ovary was stained by immunohistochemistry (**A**). The activated caspase-3 levels in the serum of NTx and Tx groups were analyzed by the ELISA assay (**B**). *Bcl2* and *cytochrome C* gene expression were analyzed by western blot (**C**,**D**). The data are representative of three independent experiments and expressed as means ± SD. * *p* < 0.05: NTx vs. Tx.

**Figure 5 antioxidants-09-00591-f005:**
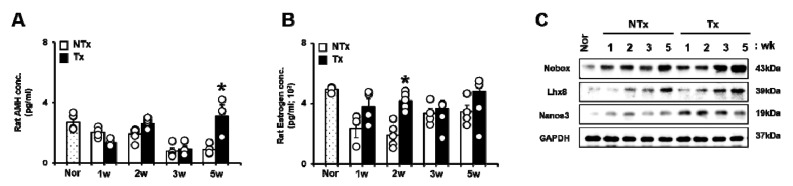
Effect of PD-MSCs transplantation on gene expression related to folliculogenesis in the ovary tissues of OVX rats. AMH and E2 hormone levels in the serum of NTx and Tx were analyzed by the ELISA assay (**A**,**B**). *Nobox*, *Lhx8*, and *Nanos3* protein expression related to folliculogenesis in NTx and Tx were analyzed by Western blot (**C**). The data are representative of three independent experiments and expressed as means ± SD. * *p* < 0.05: NTx vs. Tx.

**Figure 6 antioxidants-09-00591-f006:**
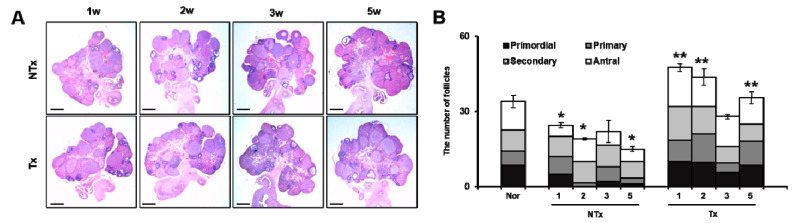
Effect of PD-MSCs transplantation on follicular development of the OVX rat model. Histopathological staining of ovary tissues was analyzed by hematoxylin and eosin staining in NTx and Tx (magnification ×10) (**A**). The numbers of total follicles and antral follicles were counted in NTx and Tx (**B**) The data are representative of three independent experiments and expressed as means ± SD. *p* < 0.05: * Normal vs. NTx, ** NTx vs. Tx.

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
