# Peer review of "Placenta-Derived Mesenchymal Stem Cells Restore the Ovary Function in an Ovariectomized Rat Model via an Antioxidant Effect"

_antioxidants, 2020, doi:10.3390/antiox9070591_

Round 1

Reviewer 1 Report

The manuscript presented by Jin Seok et al on "Placenta-derived Mesenchymal Stem Cells Restore the Ovary Function in an Ovariectomized Rat Model via Antioxidants Effect" is an interesting paper and worth merit. However with many manuscripts they require further work to bring it to a publishable level.

The main concern throughout the paper is the grammatical and typographical errors. The introduction in particular, requires the most attention as there are incomplete sentences and numerous syntax errors too numerous to list here. It is advised that the manuscript be viewed by an English editing service. There is one provided through MDPI https://www.mdpi.com/authors/english.

The authors have to be commended on the impressive diversity of techniques used in this study, however there is a concern of disjointedness of the whole study.  The link between the analysis completed and the aims are not clear. Furthermore the discussion carries this as well and only briefly touches on aspects of data interpretation. For example the exosomes in the introduction has no seeming link to the rest of the study and appears to be tacked on. 

This manuscript requires further revision particularly in the writing component to fix the wide array of errors and adjust the discussion to clearly reflect the impact of the analysis completed. There is a substantial amount of interesting and useful data here, it needs some attention to adequately represent the gravity of the study. I hope to see this manuscript corrected and published soon. Best regards. 

Reviewer 2 Report

The authors established an ovariectomized model by human placenta-derived mesenchymal stem cells (PD-MSC) transplantation conducted by intravenously injection. They found a decreased expression of oxidative stress markers and increased expression of antioxidants markers in transplantation group compared to non-transplantation group. Apoptosis were decreased in transplantation group compared to non-transplantation group. The ovary function including sex hormone and folliculogenesis were improved after PD-MSC transplantation. This is an interesting work and might be a start for mechanisms of stem cell therapy in infertility.

Author Response

Antioxidants-849767

Placenta-derived mesenchymal stem cells restore the ovary function in an ovariectomized rat model via antioxidants effect

Jin Seok, Ph.D. course; Hyeri Park. M.S. course; Jong Ho Choi, Ph.D.; Ja-Yun Lim, Ph.D., Kyung gon Kim, Ph.D., Gi Jin Kim, Ph.D

Antioxidants-849767

Reviewer #2 :

Comments and suggestions for authors:

The authors established an ovariectomized model by human placenta-derived mesenchymal stem cells (PD-MSC) transplantation conducted by intravenously injection. They found a decreased expression of oxidative stress markers and increased expression of antioxidants markers in transplantation group compared to non-transplantation group. Apoptosis were decreased in transplantation group compared to non-transplantation group. The ovary function including sex hormone and folliculogenesis were improved after PD-MSC transplantation. This is an interesting work and might be a start for mechanisms of stem cell therapy in infertility.

Author’s reply : We greatly appreciate the reviewer’s positive statement that "This is an interesting work and might be a start for mechanisms of stem cell therapy in infertility.

Reviewer 3 Report

The authors have produced a manuscript that should contribute to the advancement of knowledge in ovarian failure. In particular, how MSCs may modulate the organ function of dysfunctional disease. Furthermore, how the exosome of serum and potential therapeutic effect on ovary function by PD19 MSCs transplantation with its antioxidant capacity and oxidative stress. 

I commend the authors for some sound work in the field and their results suggest that transplanted PD-MSCs may potentially restore ovarian function. These findings may potentially offer some insights into the understanding of cell therapy for reproductive systems however, can the authors enlarge the fonts in Figure 2 (A & B) as the numbers are too small of a font size? 

Author Response

Antioxidants-849767

Placenta-derived mesenchymal stem cells restore the ovary function in an ovariectomized rat model via antioxidants effect

Jin Seok, Ph.D. course; Hyeri Park. M.S. course; Jong Ho Choi, Ph.D.; Ja-Yun Lim, Ph.D., Kyung gon Kim, Ph.D., Gi Jin Kim, Ph.D

# Reviewer 3

Comments and suggestions for authors:

The authors have produced a manuscript that should contribute to the advancement of knowledge in ovarian failure. In particular, how MSCs may modulate the organ function of dysfunctional disease. Furthermore, how the exosome of serum and potential therapeutic effect on ovary function by PD19 MSCs transplantation with its antioxidant capacity and oxidative stress.

I commend the authors for some sound work in the field and their results suggest that transplanted PD-MSCs may potentially restore ovarian function. These findings may potentially offer some insights into the understanding of cell therapy for reproductive systems however, can the authors enlarge the fonts in Figure 2 (A & B) as the numbers are too small of a font size?

Author’s reply : We greatly appreciate the reviewer’s positive statement that " These findings may potentially offer some insights into the understanding of cell therapy for reproductive systems

As Reviewer’s comments, the fonts size in Figure 2 (A & B ) were corrected.

Round 2

Reviewer 1 Report

Thank you for taking the time to improve your manuscript, particularly the discussion. The additional paragraphs and corrections has improved the flow and overall quality.

In my final read through I did notice two minor things that the authors can fix at the final proof stage.

page 2 HSP60, HSP70 and HSP90 should be included in parenthesis instead of just the numbers.

page 7 line 288 has a double period after 2.13